# Multi-scale Explainer for Graph Neural Networks

**Lutong Wu**[1]   **Shiying Cheng**[1]   **Zhiqiang Wang**[1]   **Jianqing Liang**[2]   **Peng Song**[3]   **Xizhao Luo**[4]   **Jiye Liang**[1]

## Abstract

Explainability for graph neural networks (GNNs) aims to unveil the complex decision logic of learned models by identifying the most influential structures in the input graph, thereby improving transparency and trustworthiness. Existing post-hoc explainers typically extract a sparse key subgraph at a single scale as the explanation. However, a single-scale view often fails to capture multi-level semantics, and the optimization procedure may degenerate into a local search that is sensitive to initialization and noise, leading to unstable explanations and compromising their reliability. To address these issues, we propose MSExplainer, a multi-scale explainer for GNNs. MSExplainer couples multi-scale subgraph consistency guidance with single-scale adaptive subgraph learning under a parameter-sharing design. It simultaneously extracts multi-scale key subgraphs and complementary subgraphs, yielding a hierarchical decomposition of the original graph that covers semantics at different granularities and improves the stability of subgraph extraction. Experiments on six benchmark datasets show that MSExplainer generally outperforms prior methods in explanation accuracy and fidelity. Moreover, we theoretically prove the upper bound advantage of the multi-scale strategy in representation consistency, and derive that it achieves the same-order computational complexity as single-scale methods under the parameter-sharing mechanism, thus ensuring the high fidelity of key subgraphs while maintaining computational efficiency.

[1]School of Computer and Information Technology, Shanxi University, Taiyuan, Shanxi, China [2]School of Computer Science and Engineering, Southeast University, Nanjing, Jiangsu, China [3]School of Economics and Management, Shanxi University, Taiyuan, Shanxi, China [4]School of Computer Science and Technology, Suzhou University, Suzhou, Jiangsu, China. Correspondence to: Zhiqiang Wang <wangzq@sxu.edu.cn>.

*Proceedings of the 43rd International Conference on Machine Learning*, Seoul, South Korea. PMLR 306, 2026. Copyright 2026 by the author(s).

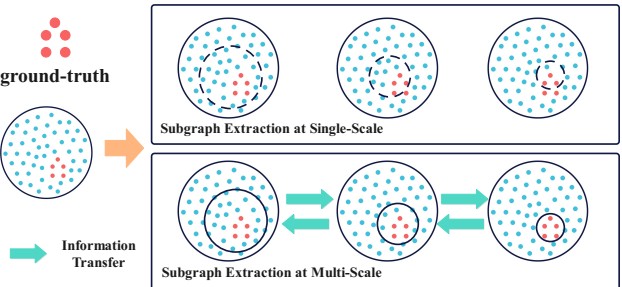

*Figure 1.* Comparison of Single-Scale and Multi-Scale Subgraph Extraction. Large scales may overlook key subgraph structures, whereas small scales are more susceptible to noise-induced deviations. Multi-scale information transfer mitigates these limitations and improves extraction stability.

## 1. Introduction

Graph neural networks (GNNs) (Ahmedt-Aristizabal et al., 2021) have achieved remarkable performance across many critical domains. However, the inherent black-box nature of GNNs has become a major barrier to reliable deployment (Fang et al., 2023; Ying et al., 2019): predictions are produced by implicitly aggregating information over the entire graph, offering no explicit rationale about which structures or features are decisive. To mitigate this issue, GNN explainability methods aim to uncover the black-box decision process by identifying the key subgraph topology and node attributes that dominate a model's prediction (Yuan et al., 2022).

Existing GNN explanation methods can be broadly categorized into instance-level and model-level approaches. Instance-level explainers (Ying et al., 2019; Yuan et al., 2021) identify input-dependent key structures for a specific sample, but typically require per-instance optimization, resulting in high computational cost and limited scalability. Model-level explainers (Yuan et al., 2020) aim to reveal global decision patterns shared across instances (e.g., class-level motifs or rules), yet may suffer from sparsity and instability due to heterogeneous graph structures.

Early methods quantify feature sensitivity via gradients or local perturbations to uncover salient substructures (e.g., GNNExplainer (Ying et al., 2019), PGM-Explainer (Vu & Thai, 2020)). Subsequent work incorporates causal reason-

ing and information-flow analysis to mitigate confounding effects and improve generalization (e.g., OrphicX (Lin et al., 2022), RC-Explainer (Wang et al., 2022), FlowX (Gui et al., 2023), MixupExplainer (Zhang et al., 2023)). More recently, to cope with complex graph semantics, explainers leverage reinforcement learning, decision-tree fitting, or graph partitioning (e.g., CiRL-Explainer (Hu et al., 2025), GraphChef (Müller et al., 2024), GSCExplainer (Wang et al., 2025)).

Despite this progress, the mainstream instance-level post-hoc paradigm still faces an optimization blind spot. Most approaches directly search for sparse explanatory subgraphs in a high-dimensional combinatorial space without global topological priors (Armgaan et al., 2024). This renders the optimization problem highly non-convex and sensitive to initialization and noise, frequently leading to local optima and entrapment by spurious correlations (Lin et al., 2022; Zhang et al., 2023). A single-scale local view further exacerbates this issue due to insufficient semantic coverage (Wang et al., 2021; Yu & Gao, 2025). As illustrated in Figure 1, a multi-scale extraction mechanism can propagate information across scales and provide feedback to dynamically adjust the learning focus at each scale, thereby better localizing key substructures.

To address the aforementioned limitations of single-scale methods and further enhance the reliability and stability of explanations, this paper proposes a Multi-Scale Explainer for GNNs (MSExplainer). This method transcends the confinement of single-granularity perspectives by constructing a hierarchical multi-scale space that captures rich semantics ranging from local details to global topology. At the core of MSExplainer are two synergistic mechanisms: multi-scale consistency guidance and single-scale adaptive learning. The integration of these mechanisms ensures that the model can stably extract high-fidelity explanatory subgraphs even in complex scenarios. The main contributions of this paper are summarized as follows:

- We propose MSExplainer, a GNN explanation method based on multi-scale decomposition. By constructing hierarchical key and complementary subgraphs and leveraging cross-scale consistency constraints to calibrate local explanation biases, MSExplainer effectively addresses the limitations of traditional single-scale methods, which are susceptible to noise interference and prone to getting trapped in local optima.

- We theoretically establish the upper-bound advantage of the multi-scale strategy in representation consistency to ensure high-fidelity explanations. Furthermore, through computational complexity analysis, we derive that MSExplainer achieves a computational complexity on the same order as single-scale explainers when employing a parameter-sharing mechanism.

- We conduct extensive experiments on six benchmark datasets, and the results show that MSExplainer outperforms existing methods on metrics such as Accuracy and Fidelity, validating its effectiveness.

## 2. Related Work

### 2.1. Instance-level Explanation

Post-hoc GNN explanation methods focus on identifying specific nodes, edges, or features that exert significant influence on model predictions, and they can be categorized into five distinct types according to recent survey research (Li et al., 2025). Mask-based methods operate by introducing learnable mask parameters over graph components to extract critical substructures that drive predictions (Ying et al., 2019; Schlichtkrull et al., 2021): For example, GN-NExplainer (Ying et al., 2019) identifies crucial nodes and edges through optimizing a soft mask that weights the importance of each graph element; K-FactExplainer (Huang et al., 2024) improves upon this framework by replacing mutual information with cross-entropy between subgraph predicted labels and ground-truth labels to avoid trivial solutions, while also aggregating outputs from multiple local explainers through a global MLP to mitigate the lossy aggregation problem. Gradient-based methods leverage backpropagated sensitivity information of model outputs with respect to input features (Smilkov et al., 2017), such as the SA method (Baldassarre & Azizpour, 2019) which calculates squared gradients to derive feature-importance scores that reflect each input's contribution to predictions. Surrogate-based methods work by fitting an interpretable proxy model in the local neighborhood of the target instance (Yuan et al., 2022); a notable example is PGM-Explainer (Vu & Thai, 2020), which constructs a probabilistic graphical model and explains predictions using a dataset generated via random perturbations, combined with the Grow–Shrink algorithm and Bayesian networks to enhance interpretability, while SubgraphX (Yuan et al., 2021) alternatively relies on Monte Carlo Tree Search and Shapley values for local subgraph exploration. Decomposition-based methods decompose final prediction scores into individual contributions from each input feature (Yuan et al., 2022); FlowX (Gui et al., 2023), for instance, builds on message-passing flows within GNNs and leverages Shapley values along with specialized sampling and learning schemes to identify the most critical multi-hop information flows that determine the model's prediction. Causal-based methods aim to discover substructures that causally drive predictions by accounting for confounding factors (Li et al., 2025): CiRLExplainer (Hu et al., 2025) constructs a causal graph to explicitly identify confounders and mitigates their influence via backdoor adjustment, then further uses reinforcement learning to generate explanatory subgraphs, thereby improving the causal validity of expla-

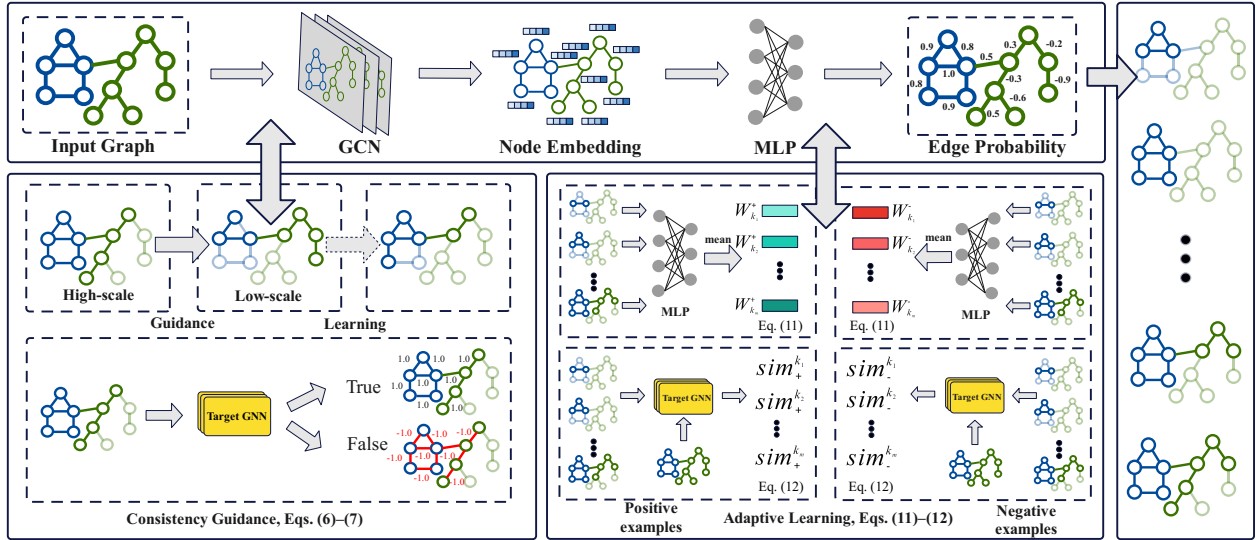

*Figure 2.* The overall framework of MSExplainer. MSExplainer first encodes node features via GCN, computes edge importance scores with an MLP, and extracts key/complementary subgraphs using multi-scale edge retention ratios. It then uses multi-scale consistency guidance to dynamically adjust edge weights, leveraging high-scale global structures to help low-scale subgraphs capture local critical patterns. Meanwhile, single-scale adaptive learning employs contrastive loss to distinguish key from complementary subgraphs within the same scale, improving the discriminability of critical structures.

nations. Despite their strong performance in achieving local explanatory accuracy, these instance-level methods typically operate under a single-scale perspective, which limits their capability to capture comprehensive semantics and maintain explanation reliability.

## 2.2. Model-level Explanation

These methods aim to uncover the global decision mechanisms of GNNs through input-agnostic, generalizable explanations that characterize knowledge patterns at the class or task level (Dai et al., 2024), typically yielding discriminative subgraph motifs or logical rules that capture structural regularities across multiple instances. For example, XGNN (Yuan et al., 2020) employs reinforcement learning to train a graph generator, optimizing representative exemplars that maximize prediction scores for the target class to produce class-level global explanations; GNNInterpreter (Wang & Shen, 2022) learns probabilistic distributions over graph structures and optimizes specific objectives to generate class-activating patterns, thereby elucidating the high-level message-passing logic underlying model decisions; GCF-Explainer (Huang et al., 2023) combines vertex-reinforced random walks with greedy summarization techniques to extract representative counterfactual graphs in the edit space, offering unified global explanations that account for both positive and negative evidence.

## 3. Problem Formulation

Given an undirected graph $G = (V, E)$, where $V = \{v_1, v_2, \ldots, v_{|V|}\}$ is the node set, the node feature matrix is $X \in \mathbb{R}^{|V| \times d}$ with $x_i \in \mathbb{R}^d$ denoting the $d$-dimensional feature vector of node $v_i$ that describes its attributes, and $E = \{e_1, e_2, \ldots, e_{|E|}\}$ is the edge set. The graph structure is represented by the adjacency matrix $A \in \mathbb{R}^{|V| \times |V|}$, where $A_{ij} = 1$ if $(v_i, v_j) \in E$, and $A_{ij} = 0$ otherwise.

This work focuses on the post-hoc explainability task for GNNs in graph classification. Let the trained target GNN be $f(\cdot)$; for an input graph $G$, the prediction is $\hat{y} = f(G)$. The target GNN $f(\cdot)$ is frozen during the explanation process. The objective of the explanation task (Wang et al., 2022) is to identify a key subgraph $G_s = (V_s, E_s)$ that plays a dominant role in the model's prediction $\hat{y}$, such that $G_s$ preserves the essential information and $f(G_s) \approx \hat{y}$.

To capture multi-level semantics, we formalize the explanation task as a multi-scale subgraph decomposition problem. Specifically, the goal is to produce a set of scale-dependent key subgraphs:

$$\mathcal{G}^* = \left\{ G_{k_m} = (V_{k_m}, E_{k_m}) \right\}_{m=1}^{M}, \tag{1}$$

where each $G_{k_m} \subseteq G$ is a subgraph of the original graph. $M$ is the preset number of scales, and each scale $m$ is associated with an edge-retention ratio threshold $k_m \in (0, 1)$ that controls the structural granularity of the subgraph.

# 4. Multi-scale Explainer

This section presents the multi-scale explanation method for GNNs, as illustrated in Figure 2.

## 4.1. Multi-scale Subgraph Extraction

Given a trained target GNN $f(\cdot)$ and an input graph $G = (V, E)$, the goal of multi-scale subgraph extraction is to compute an importance score for every edge and select the key subgraphs that contribute most to the target model's prediction at different scales. We denote the trainable shared encoder and the edge-scoring MLP of the explainer as $g_\psi(\cdot)$ and $\phi_\omega(\cdot)$, respectively, with the overall trainable parameters denoted as $\Theta = \{\psi, \omega\}$. To capture multi-level semantics while avoiding linear growth in the number of model parameters, we adopt a parameter-sharing strategy: subgraphs at different scales share the same encoder and edge-scoring MLP, and efficient subgraph generation is achieved via the following framework:

• Node representation encoding. Given the input graph $G = (V, E)$, we first extract node representations using the cross-scale shared encoder $g_\psi(\cdot)$:

$$\mathbf{H} = g_\psi(G), \qquad (2)$$

where $\mathbf{H} \in \mathbb{R}^{|V| \times d}$ represents the learned node representations, and $\mathbf{h}_i \in \mathbb{R}^d$ denotes the representation vector of node $i$. Then, for any edge $e_{ij} \in E$, we extract the feature vectors $\mathbf{h}_i$ and $\mathbf{h}_j$ of its two endpoints, compute the edge feature via element-wise multiplication, and feed it into the edge-scoring MLP $\phi_\omega(\cdot)$. The edge-importance score is defined as:

$$S(e_{ij}) = \tanh\big(\phi_\omega(\mathbf{h}_i \odot \mathbf{h}_j)\big), \qquad (3)$$

where $S(e_{ij}) \in (-1, 1)$ is the signed edge-importance score, and $\tanh(\cdot)$ maps the MLP output into a signed importance range.

• Multi-scale subgraph construction. Sort all edges in descending order of their importance scores $S(e_{ij})$, and specify a set of edge-retention ratio thresholds $\{k_m\}_{m=1}^M$. For the $m$-th scale, select the top $\lceil k_m|E| \rceil$ edges (denoted as edge set $E_{k_m}$), and collect all nodes associated with these edges (denoted as node set $V_{k_m}$). The key subgraph $G_{k_m}$ is defined as:

$$G_{k_m} = (V_{k_m}, E_{k_m}) \qquad (4)$$

Besides the key subgraph, we construct a complementary non-key subgraph using the remaining edges (i.e., the low-score edges not selected). Let $V_{k_m}^c$ denote the set of nodes associated with the remaining edges $E \setminus E_{k_m}$. The complementary subgraph is defined as:

$$G_{k_m}^c = (V_{k_m}^c, \ E \setminus E_{k_m}) \qquad (5)$$

By assigning corresponding edge-retention ratios to different scales $m$, this mechanism can systematically generate a set of key subgraphs $\{G_{k_m}\}_{m=1}^M$ and complementary subgraphs $\{G_{k_m}^c\}_{m=1}^M$ under the drive of a single set of shared explainer parameters $\Theta$.

## 4.2. Multi-scale Subgraph Consistency Guidance

In GNNs, higher-scale subgraphs contain more global information and have a higher prior probability of identifying key subgraphs. Based on this, a consistency guidance mechanism for multi-scale subgraphs is constructed: the global structural information of higher-scale subgraphs is utilized to drive the weight update of key edges, thereby guiding the accurate mining and enhancement of key subgraphs at lower scales.

This method designs an edge-importance loss $\mathcal{L}_{\text{edge}}$. For each selected edge $e_{ij} \in E_{k_m}$, the target score $t_{ij}^{(k_m)}$ is defined as:

$$t_{ij}^{(k_m)} = \begin{cases} 1, & \text{if } \arg\max\left(f(G_{k_m})\right) = \arg\max\left(f(G)\right), \\ -1, & \text{otherwise.} \end{cases} \qquad (6)$$

For the key subgraph at each scale $m$, let $S^{(k_m)}(e_{ij})$ denote the importance score of edge $e_{ij}$ when it is selected into the subgraph at scale $k_m$. We compute the $L_1$ loss between the edge-importance scores and their targets, normalize by the number of edges in that subgraph, and finally average over all scales:

$$\mathcal{L}_{\text{edge}} = \frac{1}{M} \sum_{m=1}^M \frac{1}{|E_{k_m}|} \sum_{e_{ij} \in E_{k_m}} \left| S^{(k_m)}(e_{ij}) - t_{ij}^{(k_m)} \right|, \qquad (7)$$

where $|E_{k_m}|$ is the number of edges in the subgraph at scale $m$. Given $\mathcal{L}_{\text{edge}}$, the update direction of the importance score of an edge in $G_{k_m}$ can be written as:

$$S^{(k_m)}(e_{ij}) \leftarrow S^{(k_m)}(e_{ij}) - \eta \frac{\partial \mathcal{L}_{\text{edge}}}{\partial S^{(k_m)}(e_{ij})}, \qquad (8)$$

where the gradient term is:

$$\frac{\partial \mathcal{L}_{\text{edge}}}{\partial S^{(k_m)}(e_{ij})} = \frac{1}{M|E_{k_m}|} \text{sign}\left( S^{(k_m)}(e_{ij}) - t_{ij}^{(k_m)} \right). \qquad (9)$$

When the subgraph prediction is consistent with the original graph prediction, we increase the importance scores of edges in the subgraph; otherwise, we decrease them. The low-scale subgraph is obtained by retaining the most important edges from the higher-scale subgraph, which can be viewed as a fine-grained pruning of the higher-scale subgraph. Given two scales $k_\ell$ and $k_h$ with $0 < k_\ell < k_h < 1$, we rank the edges in $E_{k_h}$ by their importance scores $S^{(k_h)}(e_{ij})$ in descending order, and define the low-scale edge set as:

$$E_{k_\ell} = \left\{ e_{ij} \in E_{k_h} \ \middle| \ \text{rank}_{k_h}(e_{ij}) \leq \left\lceil \frac{k_\ell}{k_h} |E_{k_h}| \right\rceil \right\}, \qquad (10)$$

where $\text{rank}_{k_h}(e_{ij})$ denotes the descending rank of edge $e_{ij}$ within $E_{k_h}$ according to its importance score $S^{(k_h)}(e_{ij})$. This means that the edge set $E_{k_\ell}$ of the low-scale subgraph consists of the highest-scoring edges in the high-scale edge set $E_{k_h}$, with the retained proportion determined by $k_\ell/k_h$.

This prediction-consistency-based guidance enables the low-scale subgraph to adaptively select local structures with higher discriminability during generation. Through such explicit guidance between scales, the subgraph selection progresses from global topological priors to the focus on local features.

### 4.3. Single-scale Subgraph Adaptive Learning

Single-scale subgraph adaptive learning refers to the self-adjustment of subgraphs at each scale based on their own structural information to enhance the ability of key subgraph identification. Each scale's subgraphs are not only guided by higher-scale graphs but also fine-tune edge weights by optimizing their own losses, achieving accurate localization of key subgraphs within the scale. To this end, we introduce a contrastive loss to widen the representational gap between key and complementary subgraphs at a single scale.

For each scale $m$, we compute the mean absolute edge-importance score of the key subgraph, $W_{k_m}^+$, and that of the complementary subgraph, $W_{k_m}^-$, where the absolute value reflects the influence magnitude of each edge:

$$
\begin{aligned}
W_{k_m}^+ &= \frac{1}{|E_{k_m}|} \sum_{e_{ij} \in E_{k_m}} |S(e_{ij})|, \\
W_{k_m}^- &= \frac{1}{|E \setminus E_{k_m}|} \sum_{e_{ij} \in E\setminus E_{k_m}} |S(e_{ij})|.
\end{aligned}
\tag{11}
$$

On this basis, the contrastive loss at scale $m$ is defined as:

$$
\mathcal{L}_{\text{contrastive}}^{(k_m)} = -\log \frac{\exp\left(W_{k_m}^+ \cdot \frac{\text{sim}_{k_m}^+}{\tau}\right)}{\exp\left(W_{k_m}^+ \cdot \frac{\text{sim}_{k_m}^+}{\tau}\right) + \exp\left(W_{k_m}^- \cdot \frac{\text{sim}_{k_m}^-}{\tau}\right)}, \tag{12}
$$

where $\tau$ is the temperature parameter, and $W_{k_m}^+$ and $W_{k_m}^-$ are used to adjust the contributions of the key and complementary subgraphs. The positive similarity is defined as $\text{sim}_{k_m}^+ = \cos(\boldsymbol{p}_{k_m}, \boldsymbol{p}_{\text{ori}})$, where $\boldsymbol{p}_{k_m} = f(G_{k_m})$ and $\boldsymbol{p}_{\text{ori}} = f(G)$. It measures the prediction consistency between the key subgraph and the original graph. Similarly, the negative similarity is defined as $\text{sim}_{k_m}^- = \cos(\boldsymbol{p}_{k_m}^c, \boldsymbol{p}_{\text{ori}})$, where $\boldsymbol{p}_{k_m}^c = f(G_{k_m}^c)$, and measures the prediction consistency between the complementary subgraph and the original graph. The losses over all scales are weighted by $w_m$ and summed to obtain the total contrastive loss:

$$
\mathcal{L}_{\text{contrastive}} = \sum_{m=1}^{M} w_m \mathcal{L}_{\text{contrastive}}^{(k_m)}. \tag{13}
$$

The goal of single-scale subgraph adaptive learning is to maximize the difference between the positive sample and the negative sample at each scale, so as to promote the key subgraph to be significantly separated from the complementary subgraph. The joint loss is obtained by jointly weighting the edge-importance loss and the multi-scale contrastive loss, and its formula is as follows:

$$
\mathcal{L}_{\text{total}} = \lambda_1 \mathcal{L}_{\text{edge}} + \lambda_2 \mathcal{L}_{\text{contrastive}}. \tag{14}
$$

### 4.4. Multi-scale Training

Since direct Top-K edge truncation for multi-scale subgraph extraction is discrete and non-differentiable, we use a Gumbel-Sigmoid relaxation to obtain a continuous and differentiable approximation during training. Assume the raw logit of an edge $(u, v)$ output by the edge-scoring MLP is $\alpha_{uv} = \phi_\omega(\mathbf{h}_u \odot \mathbf{h}_v)$. We generate a differentiable soft sampling mask $M_{uv}$ by introducing noise $g_{uv}$ that follows a Gumbel(0,1) distribution:

$$
M_{uv} = \sigma\left(\frac{\alpha_{uv} + g_{uv}}{\tau_g}\right), \tag{15}
$$

where $\sigma(\cdot)$ is the Sigmoid function, and $\tau_g$ is the temperature coefficient used to control the smoothness of the relaxation. When $\tau_g \to 0$, the soft mask approaches a binary edge mask; when $\tau_g$ is large, the mask becomes smoother. During training, we apply the soft mask $M_{uv}$ to the adjacency matrix or message-passing edges, allowing gradients to be back-propagated through the explainer. In the inference phase, we directly perform hard Top-K truncation based on the edge-importance scores $S(e_{uv})$ to output the final explanatory subgraph.

Under the shared-parameter multi-scale training strategy of MSExplainer (with $M$ scales and a scale ratio set $\{k_m\}_{m=1}^M$, where key subgraphs and complementary subgraphs are $G_{k_m}$ and $G_{k_m}^c$), for any input graph $G = (V, E)$, the computational cost of a single training iteration satisfies:

$$
C_{MS}(G) = C_{base}(G) + O(M|E|), \tag{16}
$$

$$
C_{single}(G) = C_{base}(G) + O(|E|). \tag{17}
$$

Therefore, when $M$ is a constant,

$$
C_{MS}(G) = \Theta(C_{single}(G)). \tag{18}
$$

Here $C_{MS}$ and $C_{single}$ denote the one-iteration cost of shared-parameter multi-scale and single-scale training, respectively; $C_{base}$ is the scale-independent shared computation, and $|E|$ is the number of graph edges. For fixed $M$, the extra cost $O(M|E|)$ is a linear traversal term with a constant coefficient, so multi-scale training has asymptotically equivalent complexity to single-scale training. The full proof is given in Appendix B.

*Table 1.* Statistics of the six datasets.

| Datasets | Mutag. | NCI1 | BA-2. | BBBP | BA-3 | MNIST-7K |
|---|---|---|---|---|---|---|
| # Graphs | 4,337 | 4,110 | 1,000 | 3,112 | 3,000 | 7,000 |
| # of classes | 2 | 2 | 2 | 2 | 3 | 10 |
| # of features | 14 | 37 | 10 | 10 | 10 | 5 |
| Avg.# of nodes | 29.75 | 29.63 | 25.00 | 26.03 | 21.92 | 81.00 |
| Avg.# of edges | 30.75 | 32.17 | 25.46 | 28.09 | 29.51 | 144.00 |

## 5. Theoretical Analysis

In this section, we analyze the advantage of the multi-scale strategy through a representation consistency theorem between the extracted key subgraph and the original graph.

**Theorem 1 (Representation Consistency of Multi-scale Strategy).** For any scale $m \in \{1, 2, \ldots, M\}$ (where $M \geq 2$ is the total number of scales), the deviation of the cosine similarity between the prediction representations of the key subgraph $G_{k_m}$ and the original graph $G$ in the embedding space satisfies the expected upper bound:

$$\mathbb{E}\left[1 - \cos\left(\boldsymbol{p}_{k_m}, \boldsymbol{p}_{\text{ori}}\right)\right] \leq O\left(\frac{1}{\ln M}\right), \qquad (19)$$

where $\boldsymbol{p}_{k_m} = f(G_{k_m})$ is the prediction distribution of the key subgraph, and $\boldsymbol{p}_{\text{ori}} = f(G)$ is the prediction distribution of the original graph (both are normalized to unit norm). The deviation is defined as $\delta_m = 1 - \cos\left(\boldsymbol{p}_{k_m}, \boldsymbol{p}_{\text{ori}}\right)$, and the expectation $\mathbb{E}[\cdot]$ is taken over the distribution of scale $m$. The upper bound $O\left(\frac{1}{\ln M}\right)$ indicates that the larger the total number of scales $M$, the lower the upper bound of the deviation. The detailed derivation can be found in Appendix A.

## 6. Experiment

### 6.1. Datasets and Setup

**Datasets.** To evaluate MSExplainer's effectiveness, we use six graph classification benchmark datasets; their statistics are summarized in Table 1.

● Mutagenicity (Kazius et al., 2005; Morris et al., 2020): It contains 4,337 molecular graphs labeled by mutagenicity, and is used for binary classification.

● NCI1 (Wale et al., 2008): It contains 4,110 molecular graphs labeled by their effects on cancer cell growth, and is used for binary classification.

● BBBP (Wu et al., 2018): It contains 2,039 molecular graphs labeled by blood–brain barrier penetration, and is used for binary classification. To alleviate class imbalance, we expand it to 3,112 graphs through sampling in our experiments.

● BA-2Motif (Luo et al., 2020): It contains 1,000 graphs,

each with a house or 5-cycle motif, and is used for binary classification.

● BA-3Motif (Yuan et al., 2022): It contains 3,000 BA-based graphs, each associated with one of three motifs, and is used for three-class classification.

● MNIST-7K (LeCun et al., 1998): It contains 7,000 superpixel-converted MNIST graphs covering 10 digit classes, and is used for ten-class classification.

**Experimental Setup.** MSExplainer is configured uniformly across all datasets. The classifier settings are as follows: a 3-layer GCN is used for BA-2Motif, BBBP, Mutagenicity, and NCI1 datasets with 2-class output; a 3-layer GCN is used for BA-3Motif with 3-class output; and a 6-layer GCN is used for MNIST with 10-class output. The explanation module consists of a 3-layer GCN and an MLP for edge-importance scoring. Nine subgraph scales are set, retaining 10%–90% of the high-importance edges in the original graph. Training uses AdamW for 300 epochs with batch size 32 and gradient clipping threshold 1. For all methods, edge masks are trained and edges are ranked, with the top-k edges selected to construct hard subgraphs, which are then fed into the target model to validate explanation quality.

### 6.2. Evaluation Metrics

**Prediction Accuracy (ACC@p):** This metric (Fang et al., 2023) is used to measure the explanation performance. It inputs the explanation subgraph into the target model and checks whether the model can recover the target prediction. It is defined as follows:

$$\text{ACC}(p) = \frac{1}{N} \sum_{i=1}^{N} \left(\mathbb{I}(\hat{y}_i = y_i)\right), \qquad (20)$$

where $N$ represents the size of the test set, $p$ represents the selection ratio, $y_i$ is the original predicted label of graph $G_i$, and $\hat{y}_i$ is the predicted label of the explanation subgraph $G_s$. The number of selected edges is $K_i = \lceil p|E_i| \rceil$, where $\lceil \cdot \rceil$ denotes the ceiling function and $|E_i|$ is the number of edges in the original graph $G_i$. Additionally, $\mathbb{I}(\hat{y}_i = y_i)$ is an indicator function that returns 1 when $y_i$ is equal to $\hat{y}_i$, and 0 otherwise.

**Explanation Accuracy with Edge AUC (Edge AUC):** This metric (Ying et al., 2019; Luo et al., 2020) is used to evaluate the edge-level accuracy of explanations. It treats edge-level explanation as a binary classification problem, where ground-truth explanatory edges are regarded as positive samples and the remaining edges as negative samples. It is defined as follows:

$$\text{EdgeAUC} = \frac{1}{N} \sum_{i=1}^{N} \text{AUROC}(\mathbf{z}_i, \mathbf{s}_i), \qquad (21)$$

where $N$ represents the size of the test set, $\mathbf{z}_i$ denotes the ground-truth edge labels of graph $G_i$, and $\mathbf{s}_i$ denotes the

Table 2. Prediction accuracy (%) of different explainer methods across datasets.

| Methods | Mutagenicity Edge Ratio | | | | | NCI1 Edge Ratio | | | | | BBBP Edge Ratio | | | | |
|---|---|---|---|---|---|---|---|---|---|---|---|---|---|---|---|
| | 0.5 | 0.6 | 0.7 | 0.8 | 0.9 | 0.5 | 0.6 | 0.7 | 0.8 | 0.9 | 0.5 | 0.6 | 0.7 | 0.8 | 0.9 |
| GNNExplainer | 65.0 | 66.6 | 66.4 | 71.0 | 78.3 | 64.2 | 65.7 | 68.6 | 75.2 | 81.8 | 48.7 | 49.0 | 49.7 | 52.9 | 66.0 |
| PGExplainer | 59.3 | 58.9 | 65.1 | 70.3 | 74.7 | 57.7 | 60.8 | 65.2 | 69.3 | 71.0 | 62.5 | 64.7 | 69.2 | 75.0 | 87.8 |
| ReFine | 63.1 | 66.4 | 68.4 | 71.4 | 84.6 | 56.0 | 66.4 | 71.8 | 77.4 | 85.2 | 51.6 | 59.6 | 64.1 | 69.2 | 75.3 |
| OrphicX | 71.4 | 71.2 | 77.2 | 78.8 | 83.2 | 66.9 | 72.7 | 77.1 | 81.3 | 85.4 | 48.4 | 47.1 | 58.0 | 64.4 | 76.0 |
| MixupExplainer | 60.6 | 61.3 | 63.8 | 71.4 | 79.3 | **72.5** | 75.4 | 78.1 | 81.0 | 88.1 | 62.3 | 70.7 | 71.2 | 71.8 | 79.2 |
| GSCExplainer | 75.1 | 78.4 | 81.1 | 83.8 | 87.2 | 70.6 | 73.5 | 78.1 | 82.1 | 84.5 | 46.2 | 50.6 | 60.9 | 73.1 | 81.7 |
| MSExplainer | **75.9** | **79.0** | **85.7** | **96.3** | **100.0** | 70.8 | **75.9** | **78.8** | **84.4** | **92.9** | **63.7** | **72.8** | **73.4** | **81.8** | **89.1** |

| Methods | BA-2Motif Edge Ratio | | | | | BA-3Motif Edge Ratio | | | | | MNIST-7K Edge Ratio | | | | |
|---|---|---|---|---|---|---|---|---|---|---|---|---|---|---|---|
| | 0.5 | 0.6 | 0.7 | 0.8 | 0.9 | 0.5 | 0.6 | 0.7 | 0.8 | 0.9 | 0.5 | 0.6 | 0.7 | 0.8 | 0.9 |
| GNNExplainer | 52.0 | 53.0 | 64.0 | 66.0 | 71.0 | 43.7 | 50.1 | 60.0 | 67.3 | 85.7 | 42.8 | 46.3 | 39.8 | 58.2 | 72.6 |
| PGExplainer | 45.0 | 46.0 | 63.0 | 76.0 | 97.0 | 41.2 | 44.5 | 48.3 | 55.6 | 79.8 | 38.6 | 41.5 | 35.7 | 52.8 | 68.9 |
| ReFine | 45.0 | 50.0 | 70.0 | 92.0 | 100.0 | 41.7 | 52.0 | 59.3 | 66.7 | 78.7 | 47.4 | 59.0 | 60.3 | 72.1 | 85.4 |
| OrphicX | 78.0 | 83.0 | 90.0 | 96.0 | 99.0 | 40.3 | 43.7 | 50.7 | 55.7 | 83.7 | 47.5 | 51.2 | 44.3 | 63.8 | 77.6 |
| MixupExplainer | 48.0 | 77.0 | 92.0 | 97.0 | 100.0 | 44.3 | 51.2 | 62.5 | 69.8 | 87.2 | 45.1 | 48.9 | 42.5 | 61.4 | 75.3 |
| GSCExplainer | 51.0 | 68.0 | 91.0 | 98.0 | 100.0 | 36.7 | 41.3 | 48.7 | 70.0 | 88.3 | 50.3 | 54.6 | 47.9 | 67.9 | 81.1 |
| MSExplainer | **84.0** | **96.0** | **98.0** | **98.0** | **100.0** | **45.7** | **52.7** | **62.5** | **73.7** | **93.3** | **55.3** | **60.1** | **67.4** | **73.3** | **86.7** |

Table 3. Explanation accuracy (%) of different explainer methods on Mutagenicity and BA-2Motif datasets.

| Dataset | MSExplainer | GSCExplainer | MixupExplainer | OrphicX | ReFine | GNNExplainer | PGExplainer |
|---|---|---|---|---|---|---|---|
| Mutagenicity | **93.0 ± 4.9** | 91.3 ± 7.7 | 61.2 ± 0.2 | 89.0 ± 2.7 | 61.2 ± 0.4 | 68.2 ± 0.9 | 83.2 ± 3.2 |
| BA-2Motif | **91.5 ± 3.2** | 88.9 ± 0.1 | 86.9 ± 0.4 | 80.4 ± 4.3 | 69.8 ± 0.1 | 64.4 ± 0.7 | 73.4 ± 11.7 |

edge importance scores generated by the explainer. A higher Edge AUC indicates more accurate edge-level explanations.

**Fidelity (Fidelity@p):** This metric (Yuan et al., 2022) is used to explore the impact on the prediction result when the important input edges identified by the explanation method are removed. It is defined as follows:

$$\text{Fidelity}(p) = \frac{1}{N} \sum_{i=1}^{N} \left( f(G_i)_{y_i} - f(\tilde{G}_i)_{y_i} \right), \quad (22)$$

where $f(G_i)_{y_i}$ represents the prediction score (e.g., confidence probability) of the target model for the original graph $G_i$ corresponding to the original predicted label $y_i$, and $f(\tilde{G}_i)_{y_i}$ represents the prediction score of the target model for the modified graph $\tilde{G}_i$ (after removing important features) corresponding to the original label $y_i$.

### 6.3. Comparison Methods

To evaluate the performance of MSExplainer, we compare it with several advanced GNN explanation methods. GN-NExplainer (Ying et al., 2019) identifies prediction-critical subgraphs by jointly optimizing trainable edge masks, while PGExplainer (Luo et al., 2020) adopts a parameterized network to learn probabilistic edge masks and can generate explanations for different graph instances after training. Re-Fine (Wang et al., 2021) defines "multi-granularity" as a hierarchy between local instance-level explanations and global class-prototype explanations, aiming to address the trade-off between class-level patterns and instance-level fidelity. OrphicX (Lin et al., 2022) is a causality-inspired method that learns causal-information-maximizing latent representations and generates architecture-agnostic explanations for arbitrary GNNs through a latent variable model. MixupExplainer (Zhang et al., 2023) enhances GNNExplainer and PGExplainer through a graph-specific Mixup framework that interpolates node features and graph structures, alleviating the distribution shift between explanatory and original subgraphs to improve explanation accuracy and generalization. GSCExplainer (Wang et al., 2025) integrates graph segmentation and contrastive learning: it divides graphs into explanatory and redundant subgraphs, generates positive

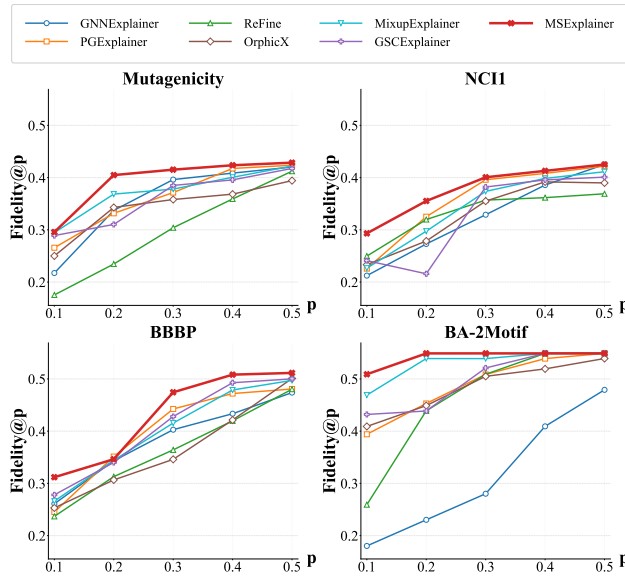

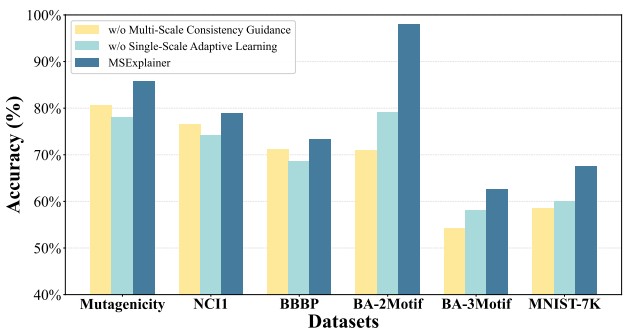

*Figure 3.* Fidelity@p comparison of different explanation methods.

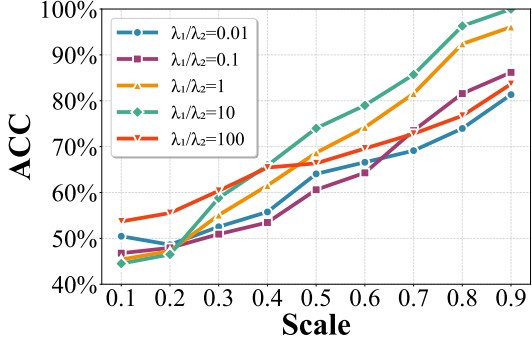

*Figure 5.* Influence of loss coefficient ratio $\lambda_1/\lambda_2$ on prediction accuracy.

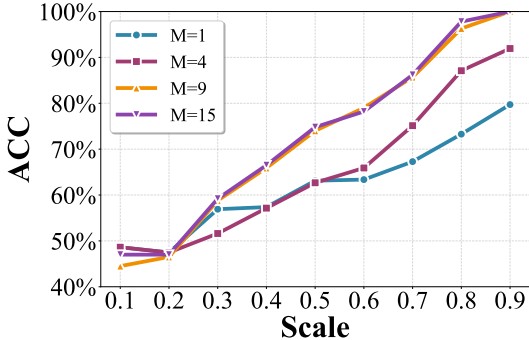

*Figure 6.* Influence of total number of scales $M$ on prediction accuracy.

respectively.

Figure 3 presents the Fidelity@p comparison of different explanation methods across four datasets. According to Eq. (22), the Fidelity metric measures the change in model predictions after removing key subgraphs at different edge ratios $p$. The results show that MSExplainer achieves higher Fidelity scores in most scenarios, demonstrating a larger drop in prediction probability when the identified key subgraphs are removed.

**Ablation Experiments.** To verify the independent contribution of the model's core components, we conducted ablation experiments on six benchmark datasets with the edge retention ratio fixed at $k_m = 0.7$. As shown in Figure 4, we compared the full model MSExplainer with two variants: w/o Consistency denotes the removal of the multi-scale consistency guidance module, retaining only the adaptive learning module; w/o Adaptive denotes the removal of the single-scale adaptive learning module, retaining only the consistency guidance module. Overall, removing either module caused the model's accuracy to decline to varying degrees across all datasets, and the variants failed to fully replicate the original model's predictions on the raw graphs. This demonstrates that relying on a single mechanism cannot ensure comprehensive explanations, and the synergy of the

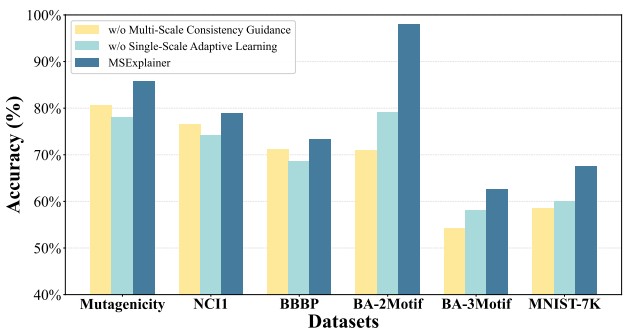

*Figure 4.* Ablation results of MSExplainer's core components across datasets.

and negative samples through edge perturbation, and optimizes a composite loss to improve explanation credibility.

### 6.4. Experimental Results

**Comparative Results.** Table 2 presents the prediction accuracy results of different explanation methods on six graph classification datasets. The prediction accuracy $\text{ACC}(p)$ is evaluated across five edge ratio scales $p$ according to Eq. (20), where the best and second-best performances are marked in bold and underlined, respectively. The prediction accuracy results demonstrate that MSExplainer achieves the highest prediction accuracy across almost all datasets and edge ratio scales.

Table 3 presents the Edge AUC scores of different explanation methods on the Mutagenicity and BA-2Motif datasets according to Eq. (21). The results show that MSExplainer achieves the highest accuracy on both datasets, outperforming the second-best method by 1.7 and 2.6 percentage points,

Original 20% 50% 70% Original 20% 50% 70%

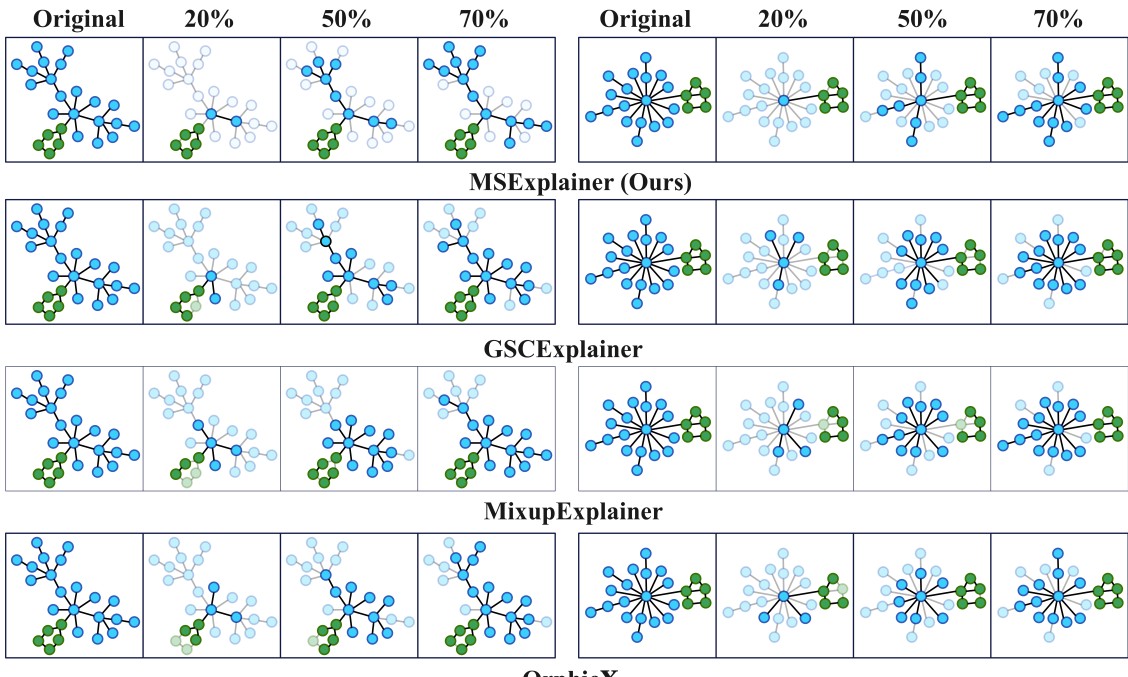

MSExplainer (Ours)

GSCExplainer

MixupExplainer

OrphicX

*Figure 7.* Visual comparison of different explanation methods on the BA-2Motif dataset. To more intuitively compare the performance of each method, one graph was selected from each of the "house" and "cycle" categories, and the edges at the 20%, 50%, and 70% scales were visualized. The ground-truth nodes are in green. In the explanation subgraphs, the key subgraphs are displayed with their original colors for prominence, while the non-key subgraphs are lightened.

dual mechanisms is critical for achieving high-performance explanations.

**Parameter Analysis.** This section investigates the impact of two key hyperparameters $\lambda_1/\lambda_2$ and $M$ on model performance. To evaluate $\lambda_1/\lambda_2$, controlled experiments were conducted with $\lambda_1/\lambda_2 \in \{0.01, 0.1, 1, 10, 100\}$ while keeping other hyperparameters (learning rate, number of epochs, batch size, etc.) fixed. As shown in Figure 5, the model achieves the best performance when $r = \lambda_1/\lambda_2 \in [1, 10]$.

For $M$, experiments on the Mutagenicity dataset were set with $M = 1$, $M = 4$, $M = 9$, and $M = 15$. Figure 6 shows that accuracy improves with increasing $M$. As the total number of scales $M$ increases, the model can capture richer multi-granularity semantics, leading to an upward trend in accuracy. However, experiments show that when $M = 9$, the model's coverage of key structures already tends to saturate, and the marginal performance gain from further increasing the number of scales diminishes gradually. To achieve optimal explanation performance while maintaining theoretical computational efficiency, this paper finally selects $M = 9$ for all experiments.

**Visual Analysis.** The visualization results in Figure 7 indicate that, compared to baseline methods, MSExplainer accurately identifies key edges within the ground-truth and ef-fectively avoids redundant edges. Particularly at low scales, MSExplainer recognizes ground-truth structures with higher stability than other models.

## 7. Conclusion

To address the limited semantic coverage and reliability limitations of existing GNN explainability methods, this paper proposes a multi-scale explainer that integrates multi-scale consistency guidance and single-scale adaptive learning to improve explanation fidelity and stability. We theoretically prove the upper-bound advantage of the multi-scale strategy in representation consistency, and experiments on six benchmark datasets confirm that MSExplainer outperforms existing methods in accuracy and fidelity, enhancing GNN transparency across diverse benchmarks. Future work will extend to directed/heterogeneous graphs and explore dynamic scale adaptation for better efficiency on large-scale graphs.

## Acknowledgements

This work is supported by the National Natural Science Foundation of China (Nos. 62272285, 62376142, U25A20529).

## Impact Statement

This paper presents work whose goal is to advance the field of Machine Learning. There are many potential societal consequences of our work, none which we feel must be specifically highlighted here.

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

# A. Proof of Theorem 1

### A.1. Definition

Given an input graph $G = (V, E)$, let the target GNN model be $f : G \to \mathbb{R}^C$, where $C$ is the number of prediction classes. MSExplainer generates key subgraphs $G_{k_m} = (V_{k_m}, E_{k_m})$ and complementary subgraphs $G_{k_m}^c = (V_{k_m}^c, E \setminus E_{k_m})$ under the multi-scale threshold set $K = \{k_1, k_2, \ldots, k_M\}$, with the thresholds assumed to be approximately uniformly distributed over $(0, 1)$. For theoretical analysis, we consider the similarity-normalized form of the contrastive objective, which focuses on the prediction-consistency term in Eq. 12 and absorbs the positive scaling factors $W_{k_m}^+$ and $W_{k_m}^-$ into the logits:

$$\widetilde{\mathcal{L}}_{\text{contrastive}}^{(k_m)} = -\log \frac{\exp\left(\cos(\boldsymbol{p}_{k_m}, \boldsymbol{p}_{\text{ori}})/\tau\right)}{\exp\left(\cos(\boldsymbol{p}_{k_m}, \boldsymbol{p}_{\text{ori}})/\tau\right) + \exp\left(\cos(\boldsymbol{p}_{k_m}^c, \boldsymbol{p}_{\text{ori}})/\tau\right)}. \tag{23}$$

This normalized form is used only for the upper-bound analysis and does not change the training objective defined in Eq. 12. Here, $\boldsymbol{p}_{\text{ori}} = f(G)$, $\boldsymbol{p}_{k_m} = f(G_{k_m})$, $\boldsymbol{p}_{k_m}^c = f(G_{k_m}^c)$, and $\tau > 0$ is the temperature parameter. The total contrastive loss is $\widetilde{\mathcal{L}}_{\text{contrastive}} = \sum_{m=1}^{M} w_m \widetilde{\mathcal{L}}_{\text{contrastive}}^{(k_m)}$. For theoretical analysis, we set $w_m = \frac{1/k_m}{\sum_{j=1}^{M} 1/k_j}$, which assigns relatively larger weights to lower-scale subgraphs.

### A.2. Model Assumptions and Preconditions

It is assumed that the GNN model $f$ satisfies Lipschitz continuity (Jia et al., 2023) : there exists a constant $L > 0$ such that $\|f(G_1) - f(G_2)\|_2 \le L d(G_1, G_2)$ (Chen et al., 2020), where $d(G_1, G_2)$ (Xu et al., 2019) is the graph edit distance:

$$d(G_{k_m}, G) = |E \setminus E_{k_m}| + \alpha |V \setminus V_{k_m}|, \alpha \ge 0. \tag{24}$$

Prediction vectors are normalized: $\|\boldsymbol{p}_{k_m}\|_2 = \|\boldsymbol{p}_{\text{ori}}\|_2 = 1$. The negative sample $\boldsymbol{p}_{k_m}^c$ follows a distribution similar to $\boldsymbol{p}_{\text{ori}}$, but structural differences lead to lower similarity. A small perturbation assumption is made: $\|\boldsymbol{p}_{k_m} - \boldsymbol{p}_{\text{ori}}\|_2 \le \epsilon < 1$. According to the theory of contrastive representation learning, the similarity-normalized objective $\widetilde{\mathcal{L}}_{\text{contrastive}}^{(k_m)}$ provides a lower bound for mutual information (Oord et al., 2018):

$$I(\boldsymbol{p}_{k_m}, \boldsymbol{p}_{\text{ori}}) \ge \mathbb{E}\left[\log \frac{\exp\left(\cos(\boldsymbol{p}_{k_m}, \boldsymbol{p}_{\text{ori}})/\tau\right)}{\exp\left(\cos(\boldsymbol{p}_{k_m}, \boldsymbol{p}_{\text{ori}})/\tau\right) + \exp\left(\cos(\boldsymbol{p}_{k_m}^c, \boldsymbol{p}_{\text{ori}})/\tau\right)}\right]. \tag{25}$$

Optimizing this loss maximizes $\cos(\boldsymbol{p}_{k_m}, \boldsymbol{p}_{\text{ori}})$, thereby minimizing the deviation $\delta_m = 1 - \cos(\boldsymbol{p}_{k_m}, \boldsymbol{p}_{\text{ori}})$.

### A.3. Deviation Analysis and Taylor Expansion Derivation

Based on Lipschitz continuity:

$$\|\boldsymbol{p}_{k_m} - \boldsymbol{p}_{\text{ori}}\|_2 \le L d(G_{k_m}, G) \le L(1 - k_m)(|E| + \alpha|V|). \tag{26}$$

For simplicity, assuming node differences are dominated by edge removal, we approximate $d(G_{k_m}, G) \le L'(1 - k_m)|E|$, where $L' = L(1 + \alpha|V|/|E|)$. Using the second-order approximation of cosine deviation for normalized prediction vectors:

$$1 - \cos(\boldsymbol{p}_{k_m}, \boldsymbol{p}_{\text{ori}}) = \frac{1}{2}\|\boldsymbol{p}_{k_m} - \boldsymbol{p}_{\text{ori}}\|_2^2 + O\left(\|\boldsymbol{p}_{k_m} - \boldsymbol{p}_{\text{ori}}\|_2^4\right), \tag{27}$$

thus, $\delta_m \le \frac{(L')^2(1-k_m)^2|E|^2}{2} + O(\epsilon^4)$. Under the small perturbation assumption, higher-order terms are neglected, so $\delta_m \le \frac{(L')^2(1-k_m)^2|E|^2}{2}$. The expected deviation is $\mathbb{E}[\delta_m] = \sum_{m=1}^{M} w_m \delta_m \le \frac{(L')^2|E|^2}{2} \sum_{m=1}^{M} w_m(1 - k_m)^2$.

### A.4. Weighted Sum Calculation and Upper Bound Refinement

For the calculation of the weighted sum: normalized weights are $w_m = \frac{1/k_m}{H_M}$, where $H_M = \sum_{j=1}^{M} 1/k_j \approx M \ln M$. Then,

$$\begin{aligned}
\sum_{m=1}^{M} w_m(1 - k_m)^2 &= \frac{1}{H_M} \sum_{m=1}^{M} \frac{(1 - k_m)^2}{k_m} \\
&\approx \frac{M \int_{1/M}^{1} \frac{(1-x)^2}{x} dx}{M \ln M} = \frac{\ln M + O(1)}{\ln M} = 1 + O\left(\frac{1}{\ln M}\right).
\end{aligned} \tag{28}$$

Thus, $\mathbb{E}[\delta_m] \leq \frac{(L')^2|E|^2}{2}\left(1 + O\left(\frac{1}{\ln M}\right)\right) = O(1) + O\left(\frac{1}{\ln M}\right)$. Considering that the optimization of contrastive loss further tightens the contribution of low thresholds, the upper bound is refined to $O\left(\frac{1}{\ln M}\right)$ (ignoring constants, with the normalization assumption $(L')^2|E|^2 = O(1)$).

### A.5. Conclusion

Compared with the single-scale method ($M = 1$ with an upper bound of $O(1)$, where the deviation does not decrease with the scale), the upper bound of the multi-scale strategy, $O\left(\frac{1}{\ln M}\right)$, decreases significantly as $M$ increases, which proves the superiority of the multi-scale approach in terms of representation consistency. This upper bound reflects the progressive advantage of the hierarchical optimization and weight mechanism, ensuring that key subgraph predictions are more faithful to the original graph.

## B. Algorithm Complexity Analysis

In the parameter sharing strategy of MSExplainer, all scales share the same encoder and scoring network, thus leading to a set of shared computational overhead that only needs to be executed once, denoted as $C_{\text{base}}(G)$. This part of the overhead includes the forward propagation of the shared graph encoder, whose complexity has a linear relationship with the number of edges $|E|$, and the constant factor is determined by the number of layers and dimensions; it also involves the edge importance evaluation that requires a single traversal of the edge set with a complexity of $\mathcal{O}(|E|)$. In addition, sorting or top-k indexing of edge importance scores is executed only once. This scale-independent overhead is incorporated into $C_{\text{base}}(G)$, so $C_{\text{base}}(G)$ remains independent of the number of scales $M$.

For each scale $m \in \{1, \ldots, M\}$, there exists an additional computational overhead with a complexity of $\mathcal{O}(|E|)$. This overhead is first reflected in subgraph construction: with the precomputed edge importance scores and their rankings or thresholds, the construction of the key subgraph edge set and the complementary subgraph edge set can be completed through linear truncation or a single traversal. Second, it includes loss calculation, which requires performing forward propagation of the message-passing GNN on both the key subgraph and the complementary subgraph separately. The cost of a single forward propagation is linear in the number of involved edges, and the total cost of processing the key and complementary subgraphs remains $\mathcal{O}(|E|)$. Summing up the additional overhead of $M$ scales, the total scale-related overhead is scale-overhead $= \mathcal{O}(M|E|)$.

The total computational cost can be compared by combining the shared overhead and the scale-related overhead. The total cost of multi-scale shared parameter training is $C_{\text{MS}}(G) = C_{\text{base}}(G) + \mathcal{O}(M|E|)$, while the total cost of single-scale training (equivalent to the case of $M = 1$) is $C_{\text{single}}(G) = C_{\text{base}}(G) + \mathcal{O}(|E|)$. When $M$ is a constant, $\mathcal{O}(M|E|) = \mathcal{O}(|E|)$, so there exist positive constants $c_1$ and $c_2$ such that for a sufficiently large $|E|$, $c_1 C_{\text{single}}(G) \leq C_{\text{MS}}(G) \leq c_2 C_{\text{single}}(G)$, that is $C_{\text{MS}}(G) = \Theta(C_{\text{single}}(G))$, which indicates that the complexity of multi-scale training is of the same order as that of single-scale training.

