# OpenReview forum: "Multi-scale Explainer for Graph Neural Networks"
_ICML.cc/2026/Conference — ICML 2026 regular_

### Official Review · Reviewer_8Jps · 2026-02-24

**Soundness:** 2
**Presentation:** 2
**Significance:** 2
**Originality:** 3
**Overall Recommendation:** 3
**Confidence:** 3

**Summary:**

This paper proposes MSExplainer, a multi-scale post-hoc explanation method for GNNs for graph classification tasks. MSExplainer applies a hierarchical multi-scale subgraph decomposition mechanism, trains a model to predict the edge importance score via multi-scale consistency guidance and single-scale adaptive learning. The authors provide a theoretical analysis and empirical experiments which demonstrate the merit of the method.

**Compliance With Llm Reviewing Policy:**

Affirmed.

**Final Justification:**

The rebuttal resolved most of my concerns, so I decide to raise the score.

**Key Questions For Authors:**

1. What is the explicit definition of scale? What is the difference between scale and sparsity?
2. What is the definition of predictive distribution $\mathbf p$ in row 238 left?
3. In the proof of theorem 1, could the authors elaborate on whether the assumptions are realistic in real-world scenarios, e.g., small perturbation assumption, normalization assumption, etc.? Could the author give any empirical evidence that the assumptions as well as the theorem hold, e.g., draw a figure of E(...) as a function of M?
5. The upper bound is not small when M is small (e.g., M = 9 in experiments). An explanation of how meaningful the theoretical decay is should be beneficial.
4. What is the definition of Edge Ratio? Is it sparsity?
5. How does the explanation look like qualitatively? Does it correspond to the ground-truths where available, e.g., BA-2motif?
6. Tab 3 shows the results of fid@0.1 and fid@0.3. A figure of fidelity as a function of sparsity would be more informative, like Fig. 4 and 5.

**Limitations:**

There's no discussion about limitations, especially a discussion about the assumptions in the theorem proof.

**Strengths And Weaknesses:**

Strength: the method is well motivated and the integration of multi-scale consistency guidance and single-scale contrastive learning is conceptually coherent.

Weakness:
1. The novelty is limited because the core components such as mask learning, contrastive loss and hierarchical guidance are not new.
2. Some definitions of symbols are missing when first used, which makes it hard to read. (see questions)
3. The difference between scale, sparsity and edge ratio is unclear, which confuses me while reading.
4. Fig2 is hard to read. For example, I don't know the meaning of W or sim and they are not explained in caption.

---

> ### Author Rebuttal · Authors · 2026-03-31
>
> Thank you for your valuable comments. We have addressed each concern as outlined below:
>
> **Regarding Weakness 1**: This paper does not claim mask learning or contrastive loss as standalone novelties. We propose a unified multi-scale synergistic optimization framework addressing single-scale limitations. By employing hierarchical optimization, our framework overcomes technical bottlenecks of conventional methods, such as noise susceptibility and local optima entrapment.
>
> **Regarding Weaknesses 2 and 3**: We will provide responses in the Key Questions.
>
> **Regarding Weakness 4**: In Fig. 2, $W_{k_m}$ is the average edge importance score in the subgraph, and $sim_{k_m}$ denotes the cosine similarity between subgraph and original prediction distributions. Definitions for $W_{k_m}^+, W_{k_m}^-, sim_{k_m}^+, sim_{k_m}^-$ are in Eq. (11). We will supplement these meanings in the revised caption to enhance readability.
>
> **Regarding Key Questions 1 & 5**: "Edge Ratio" refers to the edge retention ratio, which is explicitly controlled by the threshold $k_m$. Specifically, $k_m$ defines the proportion of top-ranked important edges retained relative to the original graph. "Scale" is defined by this threshold; each scale corresponds to a key subgraph that represents the structural granularity of the explanation. Sparsity reflects the degree of edge reduction: a lower $k_m$ results in a sparser subgraph (higher sparsity) and a lower scale. We will unify these terms in the revised manuscript for clarity.
>
> **Regarding Key Question 2**: "Predictive distribution" refers to the $C$-dimensional prediction vector from target GNN $f: \mathcal{G} \to \mathbb{R}^C$ for an input graph or subgraph. It can be intuitively understood as the GNN's output confidence distribution across all classes for the original graph or subgraph, rather than just the predicted label. We measure subgraph-original proximity by calculating the cosine similarity between their output vectors.
>
> **Regarding Key Question 3**: These assumptions are reasonable abstractions of MSExplainer’s optimization trajectory rather than universal GNN constraints.
>
> 1.Normalization: We apply unit norm processing to predictive representations, ensuring mathematical consistency for cosine similarity calculations.
>
> 2.Small Perturbation: Eq. (13) drives the subgraph representation $p_{k_m}$ toward $p_{ori}$ via cross-scale consistency. This loss-driven alignment minimizes deviation upon convergence, satisfying the small perturbation assumption in Theorem 1's proof. Analyzing the bias upper bound near the convergence point is thus well-justified.
>
> To verify Theorem 1, we provide empirical data for $\mathbb{E}[1 - \cos(p_{k_m}, p_{ori})]$ as a function of $M$ on the BA-2Motif dataset. As shown below, increasing $M$ significantly narrows the average distance between key subgraphs and the original graph in the representation space.
>
> | $M$ | 1 | 9 | 15 |
> | :---: | :---: | :---: | :---: |
> | $E\left[1-\cos\left(p_{k_m}, p_{ori}\right)\right]$ | $0.413 \pm 0.037$ | $0.212 \pm 0.021$ | $0.210 \pm 0.022$ |
>
> **Regarding Key Question 4**: $O(1/\ln M)$ describes the asymptotic trend where multi-scale yields better representation consistency than single-scale, with gains diminishing as $M$ increases. It is not intended as a precise numerical characterization of tight upper bounds for specific finite $M$. This trend corresponds to results in Figure 5, where performance gradually saturates.
>
> **Regarding Key Question 6**: To verify explanation effectiveness, we provide the following supplements:
>
> 1.Qualitative Analysis: We supplemented visualization analysis on Ba-2Motif. Results show that structures extracted by MSExplainer match ground-truth motifs (green nodes). Visualizations are available at: https://anonymous.4open.science/r/Visual-comparison/Visual_comparison.pdf
>
> 2.Quantitative Validation: We supplemented Edge AUC-ROC evaluations for datasets with ground-truth labels. As shown below, MSExplainer's accuracy in identifying core decision structures outperforms baseline methods.
>
> | DATASETS | MSExplainer | GSCExplainer | OrphicX | ReFine | GNNExplainer | PGExplainer |
> | :--- | :---: | :---: | :---: | :---: | :---: | :---: |
> |Mutagenicity|0.930±0.049|0.913±0.077|0.890±0.027|0.612±0.004|0.682±0.009|0.832±0.032|
> |BA-2Motif|0.915±0.032|0.889±0.001|0.804±0.043|0.698±0.001|0.644±0.007|0.734±0.117|
>
> **Regarding Key Question 7**:Fidelity calculation is in Eq. (20). It measures prediction change after removing top-$p$ important edges identified by the explainer. To show MSExplainer's performance across sparsity levels, we supplement Fidelity results as a function of $p$ below:
>
> |Fidelity@$p$ | 0.1 | 0.2 | 0.3 | 0.4 | 0.5 |
> | :--- | :---: | :---: | :---: | :---: | :---: |
> |NCI1|0.293| 0.355 | 0.401 | 0.413 | 0.425 |
> |BA-2motif|0.509 | 0.549 | 0.549 | 0.549 | 0.549 |
>
> Metric saturation at low sparsity indicates precise core motif localization.

---

> > ### Author Rebuttal · Reviewer_8Jps · 2026-04-03
> >
> > I decide to raise my score.

---

> > > ### Author Response · Authors · 2026-04-08
> > >
> > > Thank you for your thoughtful review. We are encouraged to know that our rebuttal addressed most of your concerns, and we sincerely appreciate the score increase. Your suggestions—particularly regarding the clarification of definitions, the empirical validation of theoretical assumptions, and the improvement of figure clarity—have been instrumental in enhancing the quality of our manuscript. We will further refine the paper based on your guidance. Thank you again for your professional assessment.

---

### Official Review · Reviewer_YreB · 2026-03-03

**Soundness:** 2
**Presentation:** 3
**Significance:** 2
**Originality:** 3
**Overall Recommendation:** 4
**Confidence:** 5

**Summary:**

In this paper, the authors propose a novel post-hoc instance-level GNN explainer, called MSExplainer, to identify influential subgraphs at multiple explanatory scales. In detail, MSExplainer comprises two main innovations, i.e., the multi-scale consistency guidance and the single-scale adaptive learning. Multi-scale consistency guidance constructs a hierarchical refinement from the high-scale explanatory subgraphs to the lower-scale ones. Within each scale, the single-scale adaptive learning adopts contrastive learning between the explanatory subgraphs and the complementary subgraphs, to consolidate accurate localization of key subgraphs. Experiments on 6 datasets demonstrate the effectiveness of MSExplainer, compared with 5 post-hoc instance-level GNN explainers.

**Compliance With Llm Reviewing Policy:**

Affirmed.

**Final Justification:**

My main concerns are resolved, therefore I'm willing to raise my rating.

**Key Questions For Authors:**

- As to the Multi-scale Subgraph Consistency Guidance, the evolution of the target score $t_{ij}^{(k_m)}$ deserves in-depth analysis. My main concern lies in the possible contradiction between different scales. For example, edge $e_{ij}$ is labeled as positive in one scale, while being labeled as negative in another scale. The contradiction above may disturb or even mislead the explainer optimization.

**Limitations:**

Possible limitations are not discussed in the current manuscript. If I miss the limitations, feel free to point it out.

**Strengths And Weaknesses:**

**Strength**

- The design principle of MSExplainer that explanations with different scales ought to be consistent is reasonable and thought-provoking. The multi-scale consistency guidance provides a practical implementation of this principle.
- The manuscript organization is well-structured and thus clearly introduces the proposed MSExplainer.

**Weakness**

- To the best of my knowledge, ReFine [1] (published in NeurIPS'21) also proposes to investigate the GNN explanation from multiple granularities (or scales), which is tightly related to the proposed methods. However, there lacks a detailed discussion about ReFine and ReFine does not serve as a baseline in order to validate the multi-scale explanation performance.
- The evaluation metrics include Prediction Accuracy and Fidelity, both of which risk the OOD issue of the target GNN model. Hence, metrics based on human-annotated ground truth, such as ROC-AUC on MUTAG, BA-2Motif, and BA-3Motif datasets, are recommended.
- The proposed MSExplainer focuses on the multi-scale explanations. However, this manuscript lacks intuitive evidence, such as edge importance visualizations, which can validate the consistency of multi-scale explanations identified by MSExplainer.
- Compared with the evaluated baselines, MSExplainer attempts to identify explanations from multiple scales, which introduces additional computational cost. The time complexity of MSExplainer is recommended to be analyzed.
- Regarding the reproducibility, the code and datasets are not available.

[1] Towards Multi-Grained Explainability for Graph Neural Networks. NeurIPS 2021.

---

> ### Author Rebuttal · Authors · 2026-03-31
>
> Thank you for your valuable comments. We have addressed each concern as outlined below:
>
> **Regarding Weakness 1**: There are fundamental differences between MSExplainer and ReFine in the definition of "scale": ReFine defines "multi-granularity" as the hierarchy between local (single instance) and global (class prototype) explanations, aiming to resolve the contradiction between class-wide patterns and instance-level fidelity. In contrast, MSExplainer defines "multi-scale" as the edge retention ratio, utilizing cross-scale consistency to ensure the explainer stably identifies core structures that drive model decisions across varying structural configurations.
>
> We include ReFine as a baseline. To ensure experimental fairness, we utilized the trained ReFine to extract explanation masks and filtered subgraphs at different scales based on weight rankings as inputs for the target model. The prediction accuracy results are as follows:
>
> | Methods / Scale | 0.5 | 0.6 | 0.7 | 0.8 | 0.9 |
> | :--- | :---: | :---: | :---: | :---: | :---: |
> | MUTAG (ReFine) | 63.1 | 66.4 | 68.4 | 71.4 | 84.6 |
> | MUTAG (MSExplainer) | 75.9 | 79.0 | 85.7 | 96.3 | 100.0 |
> | BA-2Motif (ReFine) | 45.0 | 50.0 | 70.0 | 92.0 | 100.0 |
> | BA-2Motif (MSExplainer) | 84.0 | 96.0 | 98.0 | 98.0 | 100.0 |
> | NCI1 (ReFine) | 56.0 | 66.4 | 71.8 | 77.4 | 85.2 |
> | NCI1 (MSExplainer) | 70.8 | 75.9 | 78.8 | 84.4 | 92.9 |
> | BBBP (ReFine) | 51.6 | 59.6 | 64.1 | 69.2 | 75.3 |
> | BBBP (MSExplainer) | 63.7 | 72.8 | 73.4 | 81.8 | 89.1 |
>
> **Regarding Weakness 2**: Following your suggestion, we have supplemented our evaluation with Ground-Truth-based edge AUC-ROC experiments. Results on Mutagenicity and BA-2Motif demonstrate that MSExplainer achieves significant accuracy in localizing core decision structures, outperforming current baseline methods.
>
> | Datasets / Methods  | MSExplainer | GSCExplainer | OrphicX | ReFine | GNNExplainer | PGExplainer |
> | :--- | :---: | :---: | :---: | :---: | :---: | :---: |
> |Mutagenicity|0.930±0.049|0.913±0.077|0.890±0.027|0.612±0.004|0.682±0.009|0.832±0.032|
> |BA-2Motif|0.915±0.032|0.889±0.001|0.804±0.043|0.698±0.001|0.644±0.007|0.734±0.117|
>
> **Regarding Weakness 3**: To intuitively verify the multi-scale consistency of our explanation results, we have supplemented our qualitative visualization analysis on the BA-2Motif dataset. Results demonstrate that MSExplainer maintains exceptional structural consistency across various edge retention ratios. In contrast, baseline methods often struggle to fully identify core patterns at small scales and are easily susceptible to interference from redundant structures. The corresponding visualizations are available at the following anonymous link: https://anonymous.4open.science/r/Visual-comparison/Visual_comparison.pdf
>
> **Regarding Weakness 4**: Appendix B provides time complexity analysis. With parameter sharing, training complexity is $C_{MS}(G) = C_{base}(G) + O(M|E|)$. For constant $M$, $C_{MS}(G) = \Theta(C_{single}(G))$, matching single-scale methods. We also compare actual training times on Mutagenicity across scales $M$ to demonstrate efficiency.
>
> | M | 1 | 3 | 5 | 7 | 9 |
> | :--- | :---: | :---: | :---: | :---: | :---: |
> | Training time(s) | 14.67±0.16 | 14.72±0.11 | 14.71±0.08 | 14.65±0.05 | 14.78±0.06 |
>
> **Regarding Weakness 5**:  Considering the manuscript is currently under anonymous review, complete code and datasets are provided in the supplementary materials. We will publicly release all code, data, and instructions upon official publication.
>
> **Regarding Key Question**: The aforementioned contradiction does not interfere with or mislead the explainer's optimization process. Instead, it functions as a positive supervisory signal that guides lower-scale subgraphs to identify more compact and critical structures.
>
> First, we employ a hierarchical extraction mechanism: all scales share the same set of edge importance scores $s(e_{ij})$, which are dynamically truncated based on Top-K ratios. This design ensures that in any iteration, the lower-scale subgraph $E_{k_l}$ is always a strict subset of the higher-scale subgraph $E_{k_h}$.
>
> Second, all scales share the same edge weight network (MLP) and graph encoder. This implies that the model is not training "different subgraphs" but is utilizing subgraphs of varying sizes as "test cases" to train a single scoring function $s$.
>
> Since higher-scale subgraphs retain more edges and richer global information, the high-scale parent set inherently possesses stronger reconstruction capability. Consequently, a 'conflict' only arises when the high-scale predicts correctly while the low-scale fails. This serves as the supervisory signal the model requires. Through cross-scale optimization, this design guides the explainer to accurately localize the most compact core subgraph structure within the scope defined by the high scale.

---

> > ### Author Rebuttal · Reviewer_YreB · 2026-04-03
> >
> > My concerns are basically resolved, and I decide to raise my rating.

---

> > > ### Author Response · Authors · 2026-04-08
> > >
> > > Thank you for your recognition. We are very glad that our rebuttal addressed your concerns, and we sincerely appreciate the score increase. Your suggestions—especially regarding the inclusion of the ReFine baseline and ground-truth metrics—were instrumental in enhancing the persuasiveness of our work. Thank you for your time and for acknowledging the value of our research.

---

### Official Review · Reviewer_qXwn · 2026-03-12

**Soundness:** 3
**Presentation:** 4
**Significance:** 3
**Originality:** 3
**Overall Recommendation:** 5
**Confidence:** 4

**Summary:**

This paper proposes MSExplainer, a multi-scale explainer for graph neural networks, which integrates multi-scale subgraph consistency guidance with single-scale adaptive subgraph learning to address the issues of insufficient semantic coverage and poor explainability in single-scale explanation methods. The paper theoretically proves the upper bound advantage of the proposed method in representation consistency and demonstrates that its computational complexity is of the same order as single-scale methods under the parameter-sharing mechanism. Experiments on six benchmark datasets verify that this method outperforms existing methods in explanation accuracy and fidelity.

**Compliance With Llm Reviewing Policy:**

Affirmed.

**Final Justification:**

This paper proposes a well-designed multi-scale explainer that effectively balances interpretability and computational efficiency. The method is supported by rigorous theoretical analysis and comprehensive experiments, and the paper is clearly structured and well presented.

I appreciate the authors’ thorough rebuttal, which addresses my concerns and further clarifies several aspects of the work. The responses reinforce my overall positive assessment.

**Key Questions For Authors:**

1. The method in this paper is mainly oriented to the graph classification explanation task. If extended to the node-level explanation task, will the core framework require major modifications?
2. Please elaborate on the design motivation and specific functions of the consistency constraint among multi-scale subgraphs, and how this constraint helps to improve the stability of explanation results?
3. This paper adopts a parameter-sharing mechanism for multi-scale modeling. Could you analyze the impact of this design on the generalization and training stability of the model?

**Limitations:**

Yes

**Strengths And Weaknesses:**

Strengths:
1. The multi-scale explainer proposed in this paper not only breaks through the limitations of single-scale granularity but also maintains computational efficiency comparable to single-scale methods via the parameter-sharing mechanism, achieving a balance between the explanatory capability and computational efficiency of the explainer.
2. The paper completes the theoretical proof of the upper bound of the multi-scale strategy in representation consistency and the analysis of computational complexity with rigorous logical derivation. The experimental design is compr ehensive, which fully verifies the effectiveness of the method.
3. The paper has a well-structured organization and clear logical thread. The design ideas and implementation details of the core modules are elaborated in detail, and the related work is classified and sorted out clearly. The differences between this work and existing studies are accurately distinguished, resulting in strong overall readability.

Weaknesses:
1. The edge retention ratios for multi-scale settings are manually fixed without an adaptive scale generation mechanism. It is difficult to dynamically adjust the scale division strategy according to the structural characteristics of different graph data, leading to insufficient adaptability of the method.
2. Experiments only verify the explanation performance based on the Graph Convolutional Network (GCN) model and do not extend to other mainstream GNN architectures such as Graph Attention Network (GAT) and Graph Isomorphism Network (GIN). The verification of the method's adaptability to different GNN models is inadequate.

---

> ### Author Rebuttal · Authors · 2026-03-31
>
> Thank you for your valuable comments. We have addressed each concern as outlined below:
>
> **Response to Question 1**: The core framework of MSExplainer can be adapted to node-level explanation tasks without major modifications. In node-level scenarios, the explanation target shifts to the $L$-layer computation graph $G^c$ of the target node $v$. Since MSExplainer extracts node representations via a shared encoder and calculates edge importance scores $S(e_{ij})$ using an MLP based solely on local node-pair representations, it does not involve global labels and thus possesses inherent task-agnosticism. The core consistency guidance mechanism only requires fine-tuning the graph-level consistency logic in Equation (6) to target node prediction consistency:
>
> $$ t _ {i _ j}^{(k _ m)} = \\begin{cases} 1, & \\text{if } \\arg \\max(\\hat{y} _ {v, G _ {k _ m}}) = \\arg \\max(\\hat{y} _ {v, G}) \\\\ -1, & \\text{otherwise} \\end{cases} $$
>
> where **$\\hat{y}_{v,G}$** represents the predicted label of the target node $v$ in the original graph, and **$G_{k_m}$** represents the $L$-layer computation graph of target node $v$ at the $m$-th scale. This prediction-consistency-based supervision signal can similarly guide the explainer to focus on the most critical structures at the node level.
>
> **Response to Question 2**: MSExplainer introduces multi-scale consistency guidance to address the limitations of single-scale methods, such as susceptibility to noise interference and local optima due to insufficient semantic coverage. According to Equation (10), low-scale subgraphs are strictly defined as the "fine-grained pruning" of high-scale global structures. Under this hierarchical framework, high-scale subgraphs provide macro global topological priors, while low-scale subgraphs undergo continuous calibration and "correction" of edge weights through real-time prediction feedback. In this cross-scale collaborative search, only the core edges that significantly drive model decisions across different granularities will have their importance scores $S(e_{ij})$ steadily promoted to high values through gradient backpropagation. This dynamic optimization process of continuous refinement from global to local effectively filters spurious correlations caused by random noise, ensuring the explainer can stably identify key subgraph structures.
>
> **Response to Question 3**: MSExplainer adopts a parameter-sharing mechanism to apply joint constraints across multiple retention ratios (10%–90%), significantly enhancing the model's generalization. This design essentially introduces a strong inductive bias: it compels the scoring network to identify "scale-invariant" features that maintain prediction consistency across different structural granularities, thereby effectively filtering out noise that appears only at specific scales. Regarding stability, this mechanism uses cross-scale consistency constraints to calibrate local biases in real-time, effectively preventing the explainer from falling into local optima caused by searching high-dimensional combinatorial spaces or random initialization.
>
> **Response to Weakness**: To verify the universality of MSExplainer across different mainstream GNN architectures, we have extended the scope of our experimental evaluation. We will supplement Table 2 with comparative experiments of MSExplainer on GAT and GIN architectures. Part of the experimental results are as follows:
>
> | Methods / Scale| 0.5 | 0.6 | 0.7 | 0.8 | 0.9 |
>  | :--- | :---: | :---: | :---: | :---: | :---: |
>  | **MSExplainer + GIN** | 69.35 | 73.27 | 78.57 | 84.56 | 90.78 |
>  | **MSExplainer + GAT** | 73.96 | 78.57 | 85.48 | 96.08 | 100.00 |

---

> > ### Author Rebuttal · Reviewer_qXwn · 2026-04-03
> >
> > Thanks for your detailed response. My questions are well addressed.

---

> > > ### Author Response · Authors · 2026-04-08
> > >
> > > Thank you for your acknowledgment. We are glad to know that our rebuttal and the corresponding updates have addressed your concerns. Your suggestions helped us validate the effectiveness of our model from multiple perspectives and significantly enhanced the rigor of the paper. Thank you again for your professional guidance and support for our work.

---

### Official Review · Reviewer_hvMW · 2026-03-12

**Soundness:** 4
**Presentation:** 4
**Significance:** 3
**Originality:** 3
**Overall Recommendation:** 5
**Confidence:** 4

**Summary:**

Aiming at the problems of insufficient semantic coverage and proneness to local optima in single-scale explanation methods for graph neural networks (GNNs), this paper proposes MSExplainer, a multi-scale explainer that realizes multi-scale consistency guidance and single-scale adaptive learning through hierarchical subgraph extraction. The paper theoretically proves the upper bound advantage of the proposed method in representation consistency and demonstrates that its computational complexity is comparable to that of single-scale methods. Verified on six benchmark datasets, the method outperforms existing approaches in both explanation accuracy and fidelity.

**Compliance With Llm Reviewing Policy:**

Affirmed.

**Key Questions For Authors:**

1. Could you briefly elaborate on the core logic of information transfer between high and low scales in the multi-scale strategy? A clear elaboration will make the mechanism more understandable and also confirm the integrity of the theoretical design.
2. Based on the existing experiments, could you specify the approximate range of the decline in the model’s explanation accuracy and fidelity if the guidance of high scales or low scales is removed individually?
3. Could you briefly state the most core advantage of the multi-scale strategy over the single-scale strategy in terms of subgraph extraction?

**Limitations:**

Yes

**Strengths And Weaknesses:**

Strengths:
1. The theoretical derivation is rigorous with reasonable assumptions, and the upper bound of multi-scale representation consistency is proven. The experimental design is comprehensive, with the method’s performance verified on various benchmark datasets. Meanwhile, ablation experiments and parameter analysis support the core conclusions.
2. The paper is well-structured and coherently presented, with a thorough review of the research background and related work. It clearly defines the differences from existing methods, and provides detailed descriptions of the model framework, experimental settings and other details, ensuring reproducibility.
3. This paper integrates the mechanisms of multi-scale subgraph consistency guidance and single-scale adaptive learning, and realizes multi-granularity subgraph extraction combined with a parameter-sharing design. It breaks through the limitations of the traditional single-scale explanation paradigm and effectively improves the accuracy and fidelity of model explanation.
Weaknesses:
1. The elaboration on the information transfer mechanism between scales in the multi-scale strategy is relatively brief, with an insufficient in-depth analysis of the mechanism.
2. The ablation experiments only verify the overall effect of the core modules, but do not decompose the individual contributions of high and low scales in consistency guidance, making it impossible to clarify the specific differences in the roles of different scales during the explanation process.

---

> ### Author Rebuttal · Authors · 2026-03-31
>
> Thank you for your valuable comments. We have addressed each concern as outlined below:
>
> **Response to Question 1**: The multi-scale design of MSExplainer is essentially a "global-to-local" hierarchical refinement mechanism. Structurally, low-scale subgraphs are defined as "fine-grained pruning" of high-scale subgraphs, ensuring that the edge set $E_{k_l}$ is strictly nested within $E_{k_h}$ based on the top-$k\%$ importance scores $S$.
>
> In terms of optimization logic, this mechanism is driven by two synergistic modules:
>
> Multi-scale Consistency Guidance: Implemented via edge-importance loss ($\mathcal{L}_{edge}$), it dynamically adjusts edge weights through prediction feedback to guide low-scale subgraphs in locking onto the most compact causal structures.
>
> Single-scale Adaptive Learning: Implemented via contrastive loss ($\mathcal{L}_{contrastive}$), it ensures the model identifies core features at different granularities by widening the representational gap between key and complementary subgraphs.
>
> **Response to Question 2**: According to the ablation data in Figure 3, removing either multi-scale consistency guidance or single-scale adaptive learning causes a significant drop in the model's explanation performance. To more precisely deconstruct the specific contribution of different granularities to stability, we individually shielded the guidance effect of $\mathcal{L}_{edge}$ in the high-scale range ($k_m \in [0.5, 0.9]$) and the low-scale range ($k_m \in [0.1, 0.4]$). Some ablation results are as follows:
> | Methods / Datasets | MUTAG | BA-2Motif |
> | :--- | :---: | :---: |
> | MSExplainer | 85.7 | 98.0 |
> | w/o High-scale | 85.7 | 90.0 |
> | w/o Low-scale | 79.0 | 89.0 |
>
> **Response to Question 3**: The core advantage of MSExplainer lies in solving the bottleneck where single-scale methods are susceptible to noise interference and prone to falling into local optima through multi-level optimization. This method utilizes high-scale global topological priors to filter background noise, and in coordination with low-scale nested constraints and prediction feedback, achieves dynamic calibration of edge weights.

---

### Decision · Program_Chairs · 2026-04-30

**Decision:**

Accept (regular)

**Comment:**

This paper proposes MSExplainer, a multi-scale explanation framework for graph neural networks that combines hierarchical subgraph extraction with multi-scale consistency guidance and adaptive single-scale learning. The goal is to address limitations of prior single-scale explainers, particularly in terms of semantic coverage and susceptibility to local optima.

The paper received strong overall support (1 Accept, 2 Weak Accepts, 1 Weak Reject). Notably, the reviewer who initially gave a weak reject indicates that all concerns have been fully addressed in the rebuttal, and no substantive issues remain. I therefore treat the reviews as being in overall agreement.

Reviewers consistently highlight:

* A well-motivated and carefully designed method, addressing a clear limitation in existing GNN explanation approaches.
* Theoretical support, including analysis of representation consistency and computational complexity.
* Strong empirical performance, with improvements over prior methods across multiple benchmarks.
* Clear presentation and organization.

While the contribution is primarily methodological and incremental in nature, the combination of multi-scale reasoning, theoretical grounding, and consistent empirical gains makes this a solid and useful addition to the literature.